# Influence of Water Salinity on the Growth and Survivability of Asp Larvae *Leuciscus aspius* (Linnaeus, 1758) under Controlled Conditions

**DOI:** 10.3390/ani12172299

**Published:** 2022-09-05

**Authors:** Roman Kujawa, Przemysław Piech

**Affiliations:** Department of Ichthyology and Aquaculture, Faculty of Animal Bioengineering, University of Warmia and Mazury in Olsztyn, 10-719 Olsztyn, Poland

**Keywords:** brackish water, rheophilic fish, asp, larvae, survivability, controlled conditions

## Abstract

**Simple Summary:**

Serious biological imbalances are often caused by poorly thought-out and destructive human activity, but also by progressive climate change. Each of these factors has an enormous impact on the life of terrestrial and aquatic organisms, including fish. As a result of the anthropogenically altered river environment, individual species must migrate from typically fresh to brackish waters that form in river mouths in order to conduct natural life processes such as spawning, as well as in search of food. An excessively high salinity level can be fatal to freshwater finfish, but low salinity can positively affect the growth and survival of larvae, fry and adult fish.

**Abstract:**

The effect of water with a salinity 3, 5, 7, 9 and 11 ppt on the growth and survivability of asp *Leuciscus aspius* (L.) larvae was investigated. A control sample consisted of asp larvae reared up in freshwater (0 ppt). Larvae were fed for 21 days with nauplii of the brine shrimp *Artemia salina*. Water salinity was observed to have a considerable effect on the growth and survivability of asp larvae. In addition, saline water extended the life span of *Artemia salina* nauplii, which resulted in their prolonged availability to asp larvae. Asp larvae showed low tolerance to the salinity of water, reaching 9–11 ppt. Depending on the degree of salinity, the mean final weight of larvae varied from 122.6 to 139.4 mg, at body lengths from 23.8 to 25.6 mm, respectively. The best body length increments were recorded among asp larvae maintained in water with a salinity of 3 ppt. Depending on the level of water salinity, the final survivability of asp larvae ranged from 16.9 to 94.5%. The highest and increasing mortality was demonstrated among the larvae reared in water of the salinity equal to 11 ppt. It is not recommended to rear asp larvae in water with a salinity above 7 ppt due to the low survivability and large differences in the body size of the larvae that managed to survive.

## 1. Introduction

Of all environmental factors, water salinity has an enormous impact on water-dwelling organisms. It is a significant factor that regulates the distribution of flora and fauna. In fact, water salinity limits habitats or areas where successful reproduction of a given fish species is possible. Fish inhabiting transition zones between freshwater and seawater are particularly vulnerable to changes in water salinity. Such areas include lagoons, sea bays, seaside lakes and river estuaries. Fluctuations in water salinity frequent in such water force the fish entering these zones to consume more energy for osmotic and ionic regulations, which means less energy dedicated to the growth and development [1,2]. Fish gametes, developing embryos and juvenile individuals are particularly sensitive to changes in water salinity [3,4,5]. However, the organisms most sensitive to water salinity are fish larvae in the early stage of development [6,7]. Studies on the influence of water of different salinity degrees on different development stages of fish have been conducted for many years now [8]. Most often, they have focused on fish species reared for consumption. The main purpose has been to determine the optimal level of water salinity that enables fish breeders to obtain positive effects by accelerating the growth rate of fish kept in recirculating aquaculture systems (RAS) [9,10,11,12]. Pilot studies have also been launched in order to examine the effect of salinity on the growth of fish larvae reared up for fish stocking of natural waters. An example is the study on rearing the sichel larvae *Pelecus cultratus* [7]. Such research allows us to produce more fish for stocking with less labour and energy. Among the many species raised for stocking purposes by artificial reproduction and larvae rearing is the asp *Leuciscus aspius* (L.) [13,14,15,16,17,18,19,20].

A high demand for the stocking material of this species arises from the considerably decreasing amounts of this fish in inland waters in Poland and other countries of Europe. Asp is the only fish from the Cyprinidae family which occurs in the aforementioned water bodies, where it feeds only on other fish [21]. Because of its predatory lifestyle, it plays an important regulatory role in many freshwater and brackish biocenoses. It is sometimes used as a biomanipulation tool to reduce overabundance of cyprinid fish populations, responsible in part for the increase in the eutrophication of water bodies caused by foraging on zooplankton [22,23]. Moreover, it is also a fish species that is well-liked and willingly caught by anglers in Poland and other European countries [24]. In order to keep asp populations on a satisfying level, thus meeting anglers’ expectations, it has become necessary to design protocols for the rearing of asp larvae to intensify the production of stocking material of asp and to enhance its populations in natural water bodies. While developing methods for the production of stocking material, the following factors that influence the growth and survivability of fish larvae are most often taken into consideration: type of feeds, water temperature and density of larvae [25,26,27,28,29]. In contrast, the effect of water salinity on the growth and survivability of asp larvae has not been investigated so far. Studies on various freshwater fish species reared up in water of different salinity indicate that low water salinity has a positive effect on the growth and survival rate of larvae and fish fry [7,30,31,32,33,34]. This and the aforementioned information inspired us to launch a research project with the aim to examine how water salinity affects asp larvae growth and survival.

Thus, the objective of this study was to investigate the effect of water of the salinity of 3, 5, 7, 9 and 11 ppt on the growth and survivability of asp larvae reared up under controlled conditions.

## 2. Materials and Methods

At the end of March, asp spawners were caught with gillnets from the Vistula Lagoon (Poland) waters of the salinity equal to 3 ppt. Next, the fish were transported in plastic bags filled up to 1/3 with water and up to 2/3 with oxygen to the Centre of Aquaculture and Ecological Engineering at the University of Warmia and Mazury in Olsztyn. Gametes (roe and semen) were obtained from the spawners under controlled conditions, having first hormone stimulated the fish with Ovopel [20]. The fertilized eggs were incubated for 14 days in freshwater of the temperature of 12 °C, an optimum water temperature for asp embryonic development [15]. The larvae resorbed yolk sac reserves in water at 20 °C. The larvae were reared in six separate recirculating aquaculture systems adapted to the rearing up of larvae. Asp larvae were kept in water with the salinity of 3, 5, 7, 9 and 11 ppt. The control sample consisted of larvae reared up in freshwater with no salinity (0 ppt). A single system was composed of four aquariums, with a total working capacity of 150 dm^3^. In each circulation system, there were 3 aquariums, each with a working volume of 25 dm^3^, where the larvae were reared. In order to ensure the best possible physicochemical water conditions, a fourth aquarium (in the bottom of the recirculating system) was filled up to 1/3 of the capacity with the substrate “Bactoballs” (AB Aqua Medic GmbH, Bissendorf, Germany), which served as the biological bed. Water circulation in each system was maintained with a pump by EHEIM 2280 with a maximum capacity of 1200 l h^−1^, which was coupled with an external mechanical–biological bucket filter (EHEIM GmbH & Co. KG, Deizisau, Germany). The aquariums were filled with water and the filtration system was started two weeks before asp larvae were placed in the aquariums. All aquariums were equipped with aeration cubes to maintain the best possible oxygen conditions throughout the experiment. Pro-Reef sea salt (TROPIC MARIN, Wartenberg, Germany) dissolved in tap water with a temperature of 20 °C was used to obtain the set water salinity levels. Each experimental group (salinity) was carried out in triplicate.

Asp larvae started active feeding at the age of 5 DPH (days post hatching) and measuring on average 7.8 mm in body length were counted and placed in 25 dm^3^ aquariums, achieving the density of 40 indiv.·dm^−3^. The number of larvae in each aquarium was 1000. Initially, the water temperature in the aquariums was 20 °C and was gradually (1 °C per hour) raised to 25 ± 0.1 °C, the target temperature set for the rearing up of larvae. At first, the water salinity in all aquariums (besides the control) was 3 ppt. The larvae were acclimatized to each water salinity for 3 h. The target salinity levels were obtained by gradually adding salted water to increase the salinity by 2 ppt in 3 h. During acclimatization, the larvae were not fed. Feeding was started on the next day. Once they took to swimming, the asp larvae were fed ad libitum three times a day, at four-hour intervals (8.00 a.m., 12.00 p.m. and 4.00 p.m.) The food was composed of live nauplii of the brine shrimp *Artemia salina*, which were in the first nauplii stage and size 430 microns [35]. While the asp larvae were foraging, the mobility and survivability of *Artemia* nauplii in each salinity variant were observed. The flow of water during the rearing up of larvae was maintained constant at 0.6 dm^3^·min^−1^. Throughout the entire experiment, the aquariums were illuminated with fluorescent lamps (Philips MASTER TL5 HO 39W) in the cycle of 12 h—daylight, 12 h—night. Every day, before the first feeding, the aquariums were cleansed of food debris, and dead larvae were removed and counted. The health of the larvae was monitored constantly, with the focus on the presence or absence of parasitic protozoa.

The control measurements of the body mass and total length of fish larvae were made every four days (1, 5, 9, 13 and 17 of 21 days of rearing). The first round of measurements was carried out immediately after the larvae had resorbed most of the yolk sac and started swimming but before the feeding began. Each time, samples for measurements consisted of 30 randomly caught specimens (*n* = 30) from each aquarium. They were caught in the morning (before the morning feeding). Prior to measurements, the larvae were submitted to anaesthesia using MS-222 in a dose of 50 mg·dm^−3^. Anaesthesia was carried out in water from the respective recirculation system. Sedated specimens were weighed individually on an analytical balance KERN ALJ 220-5 DNM with accuracy up to 0.1 mg (KERN & Sohn GmbH, Ballingen, Germany). At the end of measurements, larvae still under the effect of anaesthesia were placed in a Petri dish filled with water and observed under a stereoscope microscope Leica MZ16Z (Leica Microsystems GmbH, Wetzlar, Germany). The photographic documentation of larvae was produced using a microscopic camera DFC 420 (Leica Microsystems GmbH, Wetzlar, Germany). The size of larvae was analyzed with the aid of LAS V 3.1.0 software (Leica Microsystems GmbH, Wetzlar, Germany). Next, the larvae were awakened from anaesthesia and put back the aquariums from which they had been removed.

Twice daily, at 8.00 a.m. and 6.00 p.m., the water parameters, such as salinity, temperature, ammonia, nitrites and dissolved oxygen, were checked [36,37]. The water salinity measurements were obtained with a refractometer (JellyTech s.r.o., Benešov, Czech Republic). In all the groups and throughout the entire rearing period, water salinity remained on the constant, target level. Measurements of water dissolved oxygen were obtained with an oxygen meter OxyGuard Handy Polaris 2 (OxyGuard International A/S, Farum, Denmark). The oxygen content was within 7.1–7.4 mg O_2_·dm^−3^; the water pH varied from 7.2 to 7.3. No ammonia and nitrites were detected in the water during the experiment. The following were counted throughout the experiment: dead larvae during the rearing period, in a daily cycle, and the final survival rate in % expressing the ratio of fish at the end of the rearing period to the number of larvae at the onset of the experiment. The number of dead larvae recorded daily served to plot a cumulative mortality curve for every variant. Once the three-week rearing period terminated, the following were calculated for each group: the average body mass gain PM (mg) [38]:PM = mk − mp
where mk was the mean final body mass, and mp the mean initial body mass of larvae.

The index of increase in total length ITL per time unit [mm·d^−1^] was derived from the formula [39]:$$\mathrm{ITL}=\;\;\frac{TL\;\left(n_{2}\right)-TL\;\left(n_{1}\right)}{∆t}$$
where:
*TL*—mean total length of a specimen (*longitudo totalis*),*n*_1_—at the beginning of the time period,*n*_2_—at the end of the time period,∆*t*—duration of the rearing period (days—d).

The measurements were used to calculate the relative body growth rate (SGR), the index of increases in the total length per time unit (ITL), physical condition, survivability and larvae biomass. The relative specific body growth rate (SGR) and relative biomass growth rate (SBR) from the onset of feeding until the experiment was terminated were computed from the formulas [1]:$$\mathrm{SGR}=100\cdot \frac{\ln W_{2}-\ln W_{1}}{∆t},\mathrm{and}$$
$$\mathrm{SBR}=100\cdot \frac{\ln\left(n_{2}\cdot W_{2}\right)-ln\left(n_{1}\;\cdot W_{1}\right)}{∆t}$$
where
*W*_1_—mean initial weight of a reared individual (mg),*W*_2_—mean final weight of a reared individual (mg),*n*_1_—number of individuals (indiv.) at the onset of rearing,*n*_2_—number of individuals (indiv.) at the end of rearing,∆*t*—duration of the rearing-up period (days).

Next, based on the above, the relative rate of body gains (RGR) and relative rate of the biomass increase (RBR) from the onset of feeding to the termination of the experiment were computed from the following formulas [40]:$$\mathrm{RGR}=100\cdot \left(e^{\frac{SGR}{100}}-1\right),\;\mathrm{and}$$
$$\mathrm{RBR}=100\cdot \left(e^{\frac{SBR}{100}}-1\right)$$

The rates of increments (the *SGR* and *RGR*) for the body length were calculated analogously. The biomass of fish in each aquarium was determined as the product of the mean individual mass and the number of live individuals. The value thus obtained was divided by the working capacity of the aquarium, which led to the determination of the biomass of larvae in g·dm^−3^. In order to compare the results, the relative final mean length, mass and biomass of experimental fish were determined on the basis of the assumption that the length, weight and biomass of fish from the control sample (from water with 0 ppt) at the end of the experiment were 100%.

The evaluation of the normality of distribution of data and statistical processing of the results were performed with the help of the software programs Excel 2016 and Statistica 13.0 for Windows (StatSoft Inc. 2016, Tulsa, OK, USA). Statistical differences between the experimental groups were determined with the Duncan test at significance level α = 0.05 [41].

## 3. Results

There were no deaths among asp larvae observed while bringing the salinity concentration in each aquarium to the set levels. The first dead larvae were spotted a few days afterwards, regardless of the salinity of water. Their number increased as the salinity of water increased. During the 21-day rearing period, the highest survival rate was recorded in freshwater, i.e., in the control variant (0 ppt). There, 94.5% of individuals from the initial number of larvae survived the experiment (Figure 1).

A slightly worse but still high survival rate was noticed in water with a salinity of 3 ppt, where 92.5% of larvae survived to the termination of the experiment. The lowest survival rate was observed among the larvae kept in water with a salinity of 11 ppt, where only 16.9% of the initial stock lived through the rearing period. During all the control measurements, the mean lengths of larvae kept in freshwater and in water with the different salinity levels were similar (Figure 2).

On the last day of the experiment, larvae reared in freshwater achieved the mean body weight of 124.8 mg at the mean body length of 24.3 mm (Figure 3).

Slightly larger larvae grew in water with a salinity of 7 ppt. On the last day of rearing, the largest larvae were found in the aquariums filled with water with a salinity of 3 ppt, where they reached the mean body mass of 139.4 mg and mean body length of 25.6 mm. The larvae maintained in water with a salinity of 5 ppt grew to the mean weight of 134.7 mg and mean body length of 24.4 mm. Larvae reared in water with a salinity of 11 ppt achieved the final mean body weight of 126.4 mg and body length of 24.9 mm. However, none of these differences were statistically significant. The highest biomass of asp larvae in g and in g·dm^−3^ was obtained in water with a salinity of 3 ppt and in freshwater (0 ppt), where the two parameters reached, respectively, 128.95 g (5.16 g dm^−3^) and 117.94 g (4.71 g dm^−3^) (Table 1).

The lowest biomass of larvae was recorded in water with a salinity of 11 ppt, where—due to the notable mortality of larvae—it barely reached 21.36 g (0.85 g dm^−3^). Table 1 also contains the data on the other breeding parameters analyzed, such as the RGR and ITL. The results of calculations shown in Table 2 (the highest values) comparing the relative final mean length, mass and biomass of experimental fish larvae, calculated in accordance with the assumption that the length, mass and biomass of fish from the control sample (from water with 0 ppt salinity) at the end of the experiment was equal 100%, confirm that the optimal salinity of water for the rearing of asp larvae was 3 ppt. The food provided to larvae, such as *Artemia* spp. nauplii, remained alive for at least 4 h in freshwater and much longer in saline water.

## 4. Discussion

The degree of water mineralization is a significant factor that limits the settlement and spread of hydrobionts. Changes in water salinity lead to a higher consumption of energy due to the osmotic and ionic regulation, which entails less energy dedicated to the growth and development of an organism [1,42]. Most freshwater fish lose water found in tissues when placed in saline water (a hypertonic solution relative to the body fluids), which results in an increased salt concentration in the body. This can disrupt the acid–alkaline balance in the blood. If this happens, haemoglobin is unable to properly bind oxygen present in the blood, which leads to the body’s impaired oxygen management and, eventually, to death. It is likely that blood acid–base imbalance could be the cause of high mortality of asp larvae during rearing in water with a salinity of 9 and 11 ppt. A similar effect is caused by dehydration of the body. Fish larvae are particularly vulnerable to such stressors [43]. The changes mentioned above affect both larvae and the subsequent stages of fish development. Successful transfer of a freshwater fish to water with high salinity is associated with the tolerance of tissues to changes in their hydration or with the ability to regulate the osmotic pressure, e.g., through the kidneys and gills. High tolerance to water salinity, also during the larval stage of life, is demonstrated by euryhaline fish [44]. A perfect example of such fish is the Nile tilapia *Oreochromis niloticus*, which tolerates water salinity up to 20 ppt, although it is a freshwater fish [34,45,46,47]. Early stages of some other typically freshwater fish, for example, perch *Perca fluviatilis* [48], European whitefish *Coregonus lavaretus* [49] roach *Rutilus rutilus* and burbot *Lota lota* [50], sticklebacks [51] and salmonids (Salmonidae) [52], are much more tolerant to salinity than stenohaline cyprinid fish, which are more sensitive to changes in water salinity [53].

The rearing of asp larvae under controlled conditions in water with different salinity showed that water with a salinity between 3 and 7 ppt did not have a negative effect on the growth of asp larvae. The highest rearing parameters were obtained when asp larvae were maintained in water with a salinity of 3 ppt. Such results could have been expected and in fact were unsurprising because many authors, including Altinok and Grizzle, described the positive influence of low salinity water on the growth and survival of fish larvae [54]. The mentioned researchers, for example, demonstrated the positive effect of low salinity water on the fish fry of rainbow trout *Oncorhynchus mykiss* and the sturgeon subspecies Acipenser *Oxyrinchus desotoi* in water with a salinity between 3 and 9 ppt. Larvae of the perch *Perca fluviatilis* reared up in water with a salinity of 0.6–2.4 ppt also achieved a higher survival rate and more rapid growth rate than larvae of the same fish maintained in freshwater [48]. The survival rate of perch larvae in water with a salinity of 4.8 ppt ranged between 19% and 49%. It was not until the salinity of water equalled 9.6 ppt that the mortality of larvae peaked to nearly 100%.

The growing mortality of asp larvae in water with a salinity of 9–11 ppt was unsurprising. Similar observations were made on larvae of the European whitefish, a rheophilic cyprinid fish, kept in water with a salinity of 11 ppt [7]. The main reasons for high mortality might have been the consequences of total osmotic concentration and the concentration of individual ions. Another highly probable cause of the high mortality of asp larvae in water with higher salinity was the high energy expense dedicated to osmoregulation instead of the growth, a consequence of water salinity also implicated by other researchers [2]. Similar observations were reported by Altinok and Grizzle (2001), who did not report a positive effect of water salinity on the growth rate of the fish fry of stenohaline fish, such as channel catfish *Ictalurus punctatus* or the goldfish *Carasius auratus* [38,55]. If a much higher salinity concentration had been used in our experiment, the survivability of larvae would also have been affected by the availability of oxygen, whose solubility declines with increasing water salinity [55]. In our study, the concentrations of salt in water were not on a level high enough to affect the content of water dissolved oxygen. In fact, in all the experimental variants, the amounts of oxygen dissolved in water were suitable for rearing asp larvae, and no decrease in its content was noticed due to elevated water salinity. The results concerning the growth rate of asp larvae in water with different salinity are quite similar and it would be difficult to determine the impact of salinity on larvae based on these data. However, the analysis of the larval biomass obtained in each treatment allows us to discern between the optimal water salinity and the ones which had a negative effect on larvae. The essential factor is the survival rate of larvae. The larvae that survived in water with higher salinity were characterized by a more rapid growth rate. This was most probably associated with a specimen-specific resistance or tolerance to salinity.

In the course of this study, whose aim was to determine the optimal water salinity for rearing asp larvae, attention was drawn to the behaviour of live food supplied to fish larvae. The food consisted of nauplii of the brine shrimp *Artemia* spp. This is the optimal food for juvenile fish stages [56]. However, this is a seawater organism, which dies in freshwater quite quickly and subsequently sinks to the bottom, thus becoming inaccessible to fish larvae foraging in the depths of a water body. According to Sserwadda, *Artemia* nauplii die in 30 to 60 min spent in freshwater [57]. Consequently, intervals between feeding larvae with live *Artemia* spp. nauplii should be no longer than 2–3 h. In our experiment, it was noticed that nauplii of the brine shrimp died in freshwater after 4 h. The low water salinity levels tested in our experiment positively affected the viability of the brine shrimp nauplii, which was manifested by their longer life in saline water than in freshwater, making them available to foraging asp larvae for a longer time. Similar observations have been reported by other authors [32,58,59]. Water with very low salinity, of at least 2 ppt, extended the life span of *Artemia* nauplii in comparison to their longevity in freshwater. Higher salinity levels, between 7 and 9 ppt, enable brine shrimp nauplii to remain alive for up to 38 h, which has a significant impact on amounts of food supplied to fish as well as the mass of dead nauplii settling on the bottom of a fish tank and adversely affecting water quality. This consideration is especially important when fish larvae are reared in water with a temperature of 25 °C, at which decomposition of organic matter proceeds much more rapidly. Additionally, when larvae have access to live nauplii for a longer time, their body gains are faster [32].

Freshwater fish are exposed to water salinity not only when they are in a tidal zone or while migrating from freshwater to saline water bodies, but also when they stay in their natural environment. They do not need to be natural habitats between freshwater and saline water bodies, but they can be sections of a river polluted with salts. For decades, an increasing level of salinity within long sections of inland rivers has been observed due to such anthropogenic activities as mining, smelting or the chemical industry as well as the run-offs of high salinity water from stormwater canals in winter. Any influx of saline water has a considerable impact on the biocenosis of local water bodies [3,60,61], including the ichthyofauna dwelling in such waters. It is most probable that stenohaline organisms demonstrating anadrome behaviour, living mainly in rivers and often observed in waters with a salinity of a few ppt, will be best adapted to water salinity changes [62]. The results suggesting that asp larvae are able to survive and grow in water with a salinity of 3–7 ppt allow us to hypothesize that they will also survive in natural waters characterized by low and local salinity of water. As suggested by some references [60], the ide *Leuciscus idus*, a fish closely related to the asp, is able to tolerate water with a salinity as high as 15 ppt.

In conclusion, in the experiment presented above, which aimed to study the effect of water salinity on the growth and survival of asp larvae, it was found that the best growth parameter was obtained by larvae reared in water with a salinity of 3 ppt. Slightly worse, but not statistically different, a growth parameter was obtained by larvae reared in fresh water of 0 ppt and in water with a salinity of 5–7 ppt. The best survival parameter was obtained by larvae reared in water with a salinity of 0 ppt. Increasing water salinity increases the mortality of reared larvae, from 42.6% to 83.1% in water with salinities of 9 and 11 ppt, respectively.

## Figures and Tables

**Figure 1 animals-12-02299-f001:**
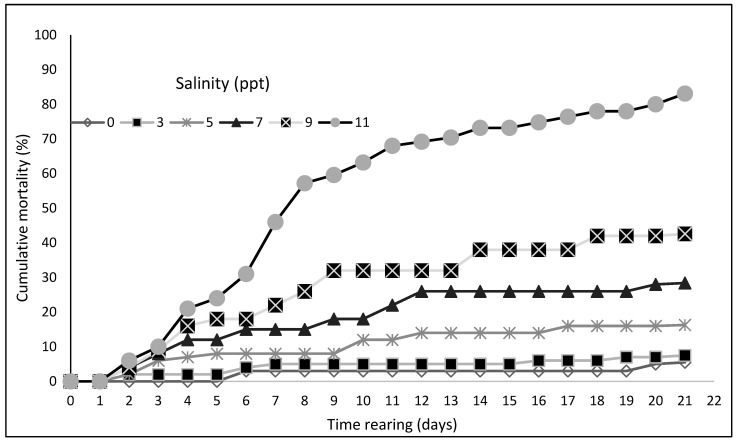
Cumulative mortality rate for asp larvae *Leuciscus aspius* (L.) reared in water with various salinity levels.

**Figure 2 animals-12-02299-f002:**
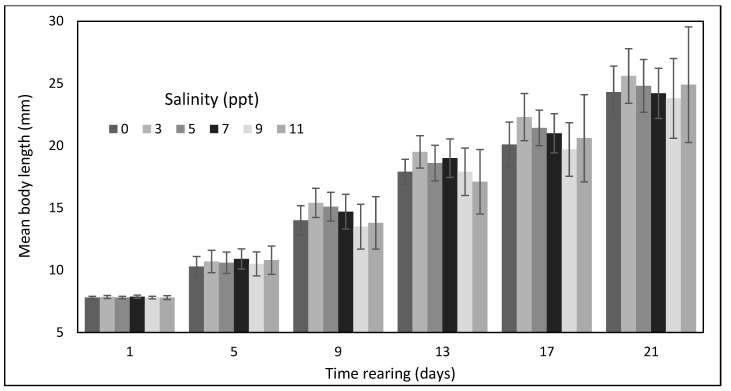
Average total body length increases in asp larvae *Leuciscus aspius* (L.) reared in water with various salinity levels.

**Figure 3 animals-12-02299-f003:**
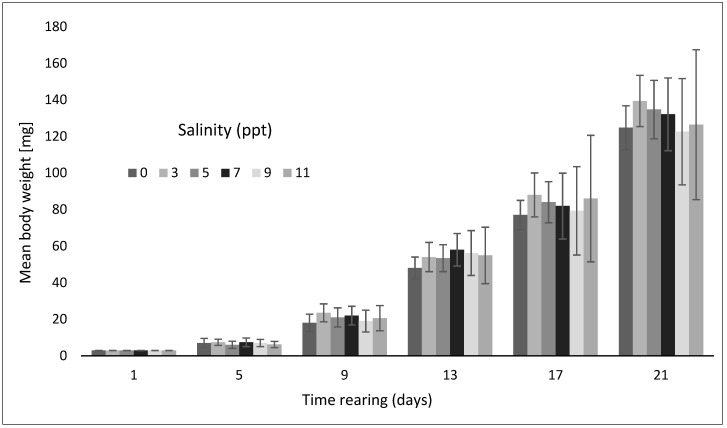
Average body weight increases in asp larvae *Leuciscus aspius* (L.) reared in water with various salinity levels.

**Table 1 animals-12-02299-t001:** Selected final parameters of asp larvae *Leuciscus aspius* (L.) rearing (mean ± SD): survival rate, increases in total length—ITL (mm·d^−1^), increases in average weight PM, larvae biomass (g·dm^−3^) are shown in parentheses (±).

Salinity (ppt)						
	0	3	5	7	9	11
Survival rate (%)	94.5 ± 3.71 ^a^	92.5 ± 4.24 ^a^	83.7 ± 5.12 ^b^	71.6 ± 3.54 ^c^	57.4 ± 4.39 ^d^	16.9 ± 3.98 ^e^
ITL (mm·d^−1^)	0.8 ± (0.02) ^a^	0.9 ± 0,02 ^b^	0.81 ± 0.02 ^c^	0.8 ± 0.02 ^a^	0.8 ± 0.03 ^a^	0.8 ± 0.04 ^a^
PM (mg)	121.9 ± 12.24 ^a^	136.5 ± 14.06 ^a^	131.8 ± 16.37 ^a^	129.2 ± 19.91 ^a^	119.7 ± 29.08 ^a^	123.5 ± 41.73 ^a^
RGR for weight (%/d)	19.62 ± 2.11 ^a^	20.25 ± 1.85 ^a^	20.06 ± 1.76 ^a^	19.94 ± 2.07 ^a^	19.52 ± 2.67 ^a^	19.69 ± 2.94 ^a^
RGR for length (%/d)	5.56 ± 0.43 ^a^	5.79 ± 0.52 ^b^	5.66 ± 0.27 ^c^	5.50 ± 0.38 ^a^	5.46 ± 0.41 ^a^	5.68 ± 0.43 ^a^
RBR for biomass (RBR) (%/d)	19.30 ± 2.43 ^a^	19.81 ± 1.03 ^a^	19.04 ± 1.14 ^a^	18.05 ± 0.42 ^ab^	16.40 ± 1.03 ^c^	9.99 ± 3.11 ^d^
Biomass (g)	117.94 ^a^	128.95 ^a^	112.74 ^a^	94.58 ^ab^	70.37 ^c^	21.36 ^d^
Biomass (g·dm^−3^)	4.7 ± 0.35 ^a^	5.2 ± 0.48 ^a^	4.5 ± 0.45 ^a^	3.8 ± 0.39 ^ab^	2.8 ± 0.28 ^c^	0.9 ± 0.26 ^d^

Results in rows with the same letter index are not statistically significantly different (α = 0.05).

**Table 2 animals-12-02299-t002:** Final relative average total length, average weight and biomass of asp larvae *Leuciscus aspius* (L.) obtained in rearing on natural feed in water with various salinity levels.

Salinity (ppt)	RFL (%)	RFW (%)	RFB (%)
3	105.35	111.70	109.33
5	102.06	107.93	95.60
7	99.59	105.85	80.20
9	97.94	98.24	59.67
11	102.47	101.28	18.11

Relative final length (RFL), weight (RFW) and biomass (RFB) of the experimental fish were calculated with the length, weight and biomass of fish in the control groups (0 ppt) at the end of the experiment assumed to be 100%.

## Data Availability

The data presented in this study are available on request from the corresponding author.

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
