# Peer review of "Influence of Water Salinity on the Growth and Survivability of Asp Larvae Leuciscus aspius (Linnaeus, 1758) under Controlled Conditions"

_animals, 2022, doi:10.3390/ani12172299_

Round 1

Reviewer 1 Report

1. Title: Influence of Water Salinity on the Growth of...  -

The authors present the results of an experiment to determine
the effect of water salinity on the growth and survival of asp larvae.
Shouldn't "survivability" be equally mentioned in the title?

2. Line 81 "waters of the salinity equal 3 ppt"

Authors should emphasize this in the Introduction section and refer to it
in the Discussion section. In described study Asp spawners came from
the salt water population (3 ppts of salinity), while the majority of the asp
population lives in freshwater. The results of rearing in saltwater larvae from
freshwater spawners could be different. Authors must also refer to this in the
Discussion section.
The authors should check whether, in the case of other species, also breeding
in brackish waters (pike, zander; Vistula Lagoon in Poland ...?), the growth rate
and survival rate of larvae reared in salt water depends/does not depend on the
origin of the spawners (fresh/salt water). If such dependencies are not clarified,
the results of the described experiment can be applied only to the larvae resulting
from the breeding of the population from saltwater.

Now another question arises: should the larvae control group be 0 ppt or maybe 3 ppt?

3. line 92 - so each experimental group (salinity) was in triplicate? You should make it clear.

4. Line 95 - AquaMedic? - The manufacturer should be listed here, not the supplier.

5. Line 97, 102, 133, 135, 137, 144 - the same as above...  (model/sample lot, manufacturer, city, country)

6. line 105 and... - check all measure units...

7. Line 120 - light: colour? (white/warm white?), temperature (Ko), light intensity?

8. Line 123 - "health of the larvae was monitored constantly" - how?

9. Line 226 - there should be no vertical lines in Table 1.

10. See also remarks in text.

Author Response

Dear Reviewer,

Thank you very much for your insightful review. All comments were taken during review process marked in the text. All main changes were marked in the text in blue.

Reviewer 2 Report

This paper reports a study on the influence of salinity on the growth of asp larvae. The results showed that salinity influenced more in survival rate than growth rate, which may supply information for aquaculture of freshwater fishes.  However, the paper presentation, especial for the discussion section, has several shortages and needs to be addressed. First, the author always simply compared their results with previous studies and found the similarity. In my opinion, it is not a way of discussion. More attentions should be payed to mechanisms and meanings related to the experimental findings. Secondly, many statements in the discussion section were different to the results, which may affect the rigorousness of the conclusion. For an example, as the results showed no remarked difference in growth, a better discussion is to explain why there was no difference rather than conclude a  'best' salinity level. 

L193-242. I cannot find anything about SGR and SBR reported in the results section.

L205-206. Fig. 2 showed that the body lengths of all groups were similar, not just those in the salinity of 3, 5 205 and 7 ppt. In addition, statistical analyses are necessary when describe those results.

L244-263. This paragraph seems like a review of literature rather than discussion. Almost nothing about the present findings were mentioned.

L265. It is not true. Actually, Table 1 showed that salinity 5 and 7 ppt reduced survival rate significantly.

L268-277. Many similar results by previous studies mean nothing. A better way is to explain the underlying mechanisms of the findings and meanings. Besides, the authors want to keep in mind that the original salinity of asp spawners was 3 PPT. Might it be a reason?

L290-291. How much oxygen would be affected by the present salinity theoretically?

L293. ‘no decrease’. However, no data was presented.

L299. You can compare the within-group variations to test if it is a possible reason. According to your idea, the variations at high salinity level should be smaller.

L320-303. So, can you talk anything about ‘the optimal water salinity for rearing asp larvae’?

L341. It is not true. The results showed that the best level for survival was 0 PPT rather than 3 PPT.

Author Response

(The authors gave the same response as above.)

Round 2

Reviewer 2 Report

I have no more copmments.